# Gender, Mental Health and Socioeconomic Differences in Fibromyalgia: A Retrospective Cohort Study Using Real-World Data from Catalonia

**DOI:** 10.3390/healthcare11040530

**Published:** 2023-02-10

**Authors:** Glòria Sauch Valmaña, Queralt Miró Catalina, Noèlia Carrasco-Querol, Josep Vidal-Alaball

**Affiliations:** 1Unitat de Suport a la Recerca de la Catalunya Central, Fundació Institut Universitari per a la Recerca a l’Atenció Primària de Salut Jordi Gol i Gurina (IDIAPJGol), 08272 Barcelona, Spain; 2Health Promotion in Rural Areas Research Group, Gerència Territorial de la Catalunya Central, Institut Català de la Salut, 08272 Barcelona, Spain; 3Unitat de Suport a la Recerca Terres de l’Ebre, Fundació Institut Universitari per a la Recerca a l’Atenció Primària de Salut Jordi Gol i Gurina (IDIAPJGol), 08272 Barcelona, Spain; 4Faculty of Medicine, University of Vic-Central University of Catalonia, 08500 Vic, Spain

**Keywords:** anxiety, depression, fibromyalgia, men, women

## Abstract

The main objective of our study was to assess the associated risk between fibromyalgia (FM) and the incidence of the diagnosis of anxiety and depression in the general population during the years 2010–2017 in Catalonia. Method: A retrospective cohort study was designed using the Information System for Research Development in Primary Care database. All patients with FM were included (n = 56,098) and matched to the control group in a 1:2 pairing ratio (n = 112,196). The demographic variables studied were sex, age and socio-economic status. Results: Patients with FM have a lower survival rate if they are also diagnosed with anxiety and depression during the entire study period, with the rate being 26.6% lower in FM patients at an 8-year follow-up (0.58, 95%CI: 0.57–0.59 vs. 0.79, 95%CI: 0.78–0.79). There is a 58% reduction in the risk of developing anxiety and/or depression in the control group vs. the FM group (*p*-value < 0.05), and by 45% in male vs. female sex (*p*-value < 0.05). Conclusions: FM is a disease that is associated with anxiety and depression, and men are at lower risk of anxiety and depression after FM diagnosis.

## 1. Introduction

FM is a disease of unknown and complex aetiopathogenesis, and although pain is its main characteristic, it also stands out for being a disease with multiple symptoms such as fatigue, sleep disorders, anxiety and depressive disorders. For thirty years, various diagnostic criteria, screening and instruments have been developed in the clinical setting, posing a major challenge to researchers and clinicians worldwide. However, the most widely used criterion has been that of the American College of Rheumatology, 1990 [1], as revised in 2016 [2]. Some studies point to genetic predisposition, environmental triggers and neuromodulation being involved in the onset and course of the disease [3,4], while other studies point to the existence of predisposing factors [5] being associated with a high rate of functional disability [6].

Since 2016 in Catalonia, 18 specialised Central Sensitivity Units (USCSS) formed by interdisciplinary teams based both in hospitals and primary care centres have been accredited, aiming to provide care to patients with central sensitization problems.

FM affects between 0.2 and 6.6% of the world’s population and is more prevalent in middle-aged women. Men also suffer from the disease, but they experience different facets of pain and fatigue compared to women; the pain is more localised as opposed to women, who manifest more generalised pain [7]. A literature review found that the symptoms experienced by men could not be identified in the same way as in women, so there could be an underdiagnosis of this disease in men [5], who manifest more sleep pattern disturbances [8] and a poorer quality of life [9]. Men may also be less likely to seek medical help for FM symptoms, which may result in delayed diagnosis and treatment.

There is a lack of literature regarding the comorbidities of fibromyalgia based on population-based data. Anxiety and depression are two comorbidities frequently associated with fibromyalgia and chronic pain, being present in 30–50% of patients at the time of disease diagnosis. It can be stated that patients with FM suffer more anxiety and depression than those without the disease [10]. The prevalence of depression in patients with FM is estimated to be between 18 and 36% and between 11.6 and 32.2% for anxiety disorders. In addition, some studies suggest that certain personality traits are associated with anxiety and depression [11]. Likewise, FM pain and functional disability can increase depression and anxiety [12], and suicidal ideation has even been shown to be a significant parameter in patients with chronic pain, attributing it to the suffering, depression and anxiety they experience [13]. Studies have also found that patients with fibromyalgia and depression have a poorer quality of life, greater disability and higher health care costs than patients with only one of these disorders [14].

The aim of this population-based study was to compare the occurrence of anxiety and depression in patients diagnosed with FM and patients without FM during the years 2010–2017 in Catalonia.

## 2. Materials and Methods

A retrospective cohort study was designed between 1 January 2010 and 31 December 2017 in Catalonia. The scope of the study was the 283 primary care teams (EAP) managed by the Catalan Health Institute (ICS). Patient data were obtained from the medical records of the information system for the development of research in primary care (SIDIAP) [15]. It is a clinical database with pseudo-anonymised data used for the ICS electronic health record system in primary care (ECAP) and other complementary data sources containing individual patient information linked to a unique and anonymous identifier [16,17]. This database contains 6,378,910 (80% of the population of Catalonia) persons aged 18 years or older. The information collected includes socio-demographic and pharmacological prescription information, as well as clinical measures and diagnoses collected using the International Classification of Diseases (ICD 10), among others. 

To obtain the total number of people with FM for this study, we excluded people who did not have all the socio-demographic variables (sex, MedeA, among others), and the control group was obtained by pairing on a 1:2 case-control ratio with the same age, sex and primary care team. In total, and after excluding patients with anxiety and depression, we obtained a sample of 101,178 patients: 20,968 patients with FM and 80,210 patients in the general population control group without FM. People with FM are a mix of prevalent cases (7641 are treated as if they got their diagnosis on 1 January 2010) and incident cases (13,327) during the period of study.

The demographic variables studied were sex, age and information on socio-economic status recorded through the MedeA index, which detects small areas in large cities with unfavourable socioeconomic characteristics associated with mortality. This is a deprivation index linked to each residential census tract in the population. This index is only available for urban areas, defined as municipalities with more than 10,000 inhabitants and a population density of more than 150 inhabitants/km^2^. The rest of the areas were considered rural. This deprivation index is classified into quintiles, from MedeA 1 (low deprivation) to MedeA 5 (high deprivation). As a composite deprivation index, it assesses barriers to accessing employment, education, culture, and social development at a level that is considered acceptable to the society or surrounding region, and it is composed of subindicators for employment and education [18].

Anxiety and depressive disorders were identified on the basis of the International Statistical Classification of Diseases and Related Health Problems, Tenth Revision, Clinical Modification (ICD-10-CM) of interest. Codes commonly used in clinical practice were F32.9 for depression (major depressive disorder, single episode) and F41.1.1 for anxiety (generalised anxiety disorder).

Both groups (cases N = 20,968 and controls N = 80,210) of participants were free of anxiety and depression at baseline (2010). All outcomes were determined during the study period (2010–2017) and identified with the ICD-10 code. 

This study was approved by the IDIAP Jordi Gol’s Research Ethics Committee (Code P18/081). The data was anonymised and processed with the utmost confidentiality.

### Statistical Analysis

The baseline characteristics of the patients were described by descriptive analysis: we expressed continuous variables as mean (standard deviation) or median (1r–3r quantile), and we summarised categorical variables as absolute frequency (percentage). For bivariate analyses, we used the chi-square test for categorical variables and the Student’s *t*-test or Mann whitney for numerical variables. In order to describe and compare the time to onset of anxiety and/or depression between different groups, the Kaplan–Meier method was calculated as of 1 January 2010 for prevalent cases, and for incident cases, it was calculated on actual diagnostic dates. We first performed survival analysis using Kaplan–Meier curves separating patients according to general population or FM population and sex and then estimated a Cox model to model the risk of depression/anxiety, considering all study variables as confounding factors: sex, age and MedeA. The sensitivity analysis was performed to validate whether the analyses of cases together (prevalent and incident) were valid. The analyses were performed with version 4.1.2 of the R software, the confidence intervals were 95% and the significance level was set at 0.05.

## 3. Results

The demographic and baseline characteristics are described in Table 1. The majority are women (94.5%) of middle age (53.0 years old), living in urban areas, who do not consume alcohol in a way that puts their health at risk and are not smokers.

Table 1 shows the demographic characteristics of the two groups and profiles of cases: incident or prevalent. 

Of the FM group, 32.3% developed anxiety and/or depression between 2010 and 2017, more than double that of the control group (*p*-value < 0.05). With respect to the control group, twice as many FM patients developed only anxiety (*p*-value < 0.05) and more than three times as many developed only depression (*p*-value < 0.05), Table 2. N = 20,968 corresponds to FM cases without anxiety and depression and N = 80,210 refers to FM control without anxiety and depression.

Kaplan–Meier survival curves for the outcomes are depicted in Figure 1 and Figure 2. Patients with FM have a lower survival rate if they are also diagnosed with anxiety and/or depression during the entire study period; their survival rate was 26.6% lower in FM patients at 8 years of follow-up (0.58, 95%CI: 0.57–0.59 vs. 0.79, 95%CI: 0.78–0.79), Figure 1.

This difference, during the entire period, is observed both in women (survival of 0.57 at eight years of follow-up, 95%CI: 0.56–0.58 in the FM group vs. 0.89, 95%CI: 0.78–0.79 in the general population) and in men (survival of 0.67 at an 8-year follow-up, 95%CI: 0.64–0.71 in FM group vs. 0.89, 95%CI: 0.88–0.90 in the general population), Figure 2.

To study the onset time of anxiety and/or depression considering the variables studied individually, Cox regression was used (Table 3). There is a 58% lower risk of developing anxiety and/or depression in the control group vs. the FM group (*p*-value < 0.05). There is also a lower risk among males vs. females (45% less, *p*-value < 0.05). No differences were observed according to age or MedeA level.

## 4. Discussion

The purpose of this study was to compare the risk of suffering from anxiety and/or depression in women and men diagnosed with FM and women and men without FM during the years 2010–2017 in Catalonia (Southern Europe). Significant impact differences have been found in previous studies, at the European level, between the southern area versus the northern and central areas of Europe. Higher impact scores were observed in women with fibromyalgia from the southern area of Europe [19].

Although the cause of FM is not yet fully understood, central sensitisation of the central nervous system, which is associated with the development of increased sensitivity to pain signals, is the most widely accepted mechanism by experts [20]. FM is a syndrome that can severely alter the quality of life, including physical and mental fatigue, among others. Mood disorders have been associated with a negative impact on pain, sleep, fatigue and overall quality of life. The most common psychological problems in FM patients are anxiety and depression, with prevalence rates of 18–36% for depression and 11.6–32.2% for anxiety disorders [21,22]. Some studies show that affective temperament traits are subclinical manifestations of mood disorders [23]. 

The results of our study indicate that psychiatric disorders, specifically anxiety and depression, are highly associated with fibromyalgia in the Catalonian population. Their onset worsens the progress of the disease and makes it chronic [24]. Anxiety is the most prevalent condition in our fibromyalgia patients compared to the control group. These high levels of anxiety have also been observed in other studies [12]. Although antidepressant treatments and psychological therapies are currently used to improve the quality of life in these patients, none of the drugs currently available are sufficient to treat the main symptoms of FM.

In our study, we highlight the fact that people diagnosed with FM are at higher risk of anxiety and depression compared to the control group, and women are at higher risk of anxiety and depression at diagnosis than men. This is consistent with the other studies reported in the literature, in which anxiety and depression is more prevalent in females [25].

When patients with FM are compared with healthy controls, a significantly higher prevalence of mood disorders has been reported [26]. This is consistent with our study, since among the cases we found that 32.3% of FM patients were diagnosed with anxiety and depression, while just 14.5% of the control group were diagnosed with anxiety and depression. Anxiety amongst the FM patients was 21.1%, twice higher than in the control group, and depression was 19%, more than three times higher than in the control group. These results coincide with some studies comparing FM patients with healthy control groups, showing that the prevalence of mental disorders is significantly higher in patients with FM [27,28]. 

When analysed by sex, the percentage of women with depression and anxiety was reported to be higher than that of men. This may be due to the fact that, in the general population, men have a lower rate of depression and anxiety diagnoses [29]. However, in contrast to our results, some other studies highlight a high frequency of psychiatric disorders in men with FM [30], which is contrary to what happens in the general population, where women present depression and anxiety disorders more frequently than men [31].

In addition, in the 8-year follow-up, the probability of developing anxiety or depression was twice as high in cases (0.42) than in control groups (0.21) since the diagnosis of FM (Table 3). 

Furthermore, one study also found that the severity of depression and/or the presence of bipolar trait symptoms parallels the severity of FM [32]. The quality of life of people with FM appears to be a reflection of pain, anxiety and depression; these three factors interact in a loop and can feed back on each other. Moreover, there are studies that link depression to quality sleep, with 80% of patients reporting poor sleep [33]. Poor sleep quality is a prominent symptom, and the main sleep problems described for this condition include reduced sleep efficiency [34]. However, the association between pain and sleep is complicated, and the direction of these relationships is not yet well established [35].

Taking into account the limitations of drug treatment in FM and the potential of multicomponent non pharmacological therapies, it’s interesting to consider this last approach, which includes a psychological work to improve the life quality of these patients. 

FM is more prevalent in women; this data is consistent with all studies reported in the literature and the one we have shown in our latest work [36]. The time to diagnosis is usually about 6 years, and comorbidity, patient age (older), and lack of experience of the physician have been associated with a longer time to diagnosis [37]. This last point needs to be improved through physician formation in the matter and the support of the Central Sensibilisation Units of ICS Catalonian Primary Care.

The limitations of our study deserve to be considered. First, all data and diagnoses were extracted from the SIDIAP electronic database, so we cannot exclude the possibility that a diagnosis may have been underestimated in some patients with FM, i.e., that the corresponding diagnostic label was not given. We have analysed prevalent and incident cases together because the information on the date of diagnosis from the computerised clinical history was not updated and reliable until 2010. In order not to lose sample size and for this to be more representative, we analysed all cases, taking into account that the date of diagnosis for prevalent cases was 1 January 2010.

Furthermore, the SIDIAP database only has 75% of the population living in Catalonia; the other 25% of the population is distributed among other providers. Although the data collection in the SIDIAP database has been previously validated, there is still a possibility of misclassification [16]. 

However, our study has some strengths, such as the large sample size and broad representativeness. In addition, with the cohort design, we have been able to evaluate the differences between variables.

## 5. Conclusions

FM is a disease that is associated with anxiety and depression, although men are at lower risk of anxiety and depression at diagnosis. The trend toward the biopsychosocial model highlights the importance of psychopathology and early referral to mental health professionals to improve the quality of life of patients and guide the development of individualised multicomponent nonpharmacological therapies according to gender differences. Our findings point to the importance of FM to the public health system and the need to prepare public policies and provide adequate care to improve the daily lives of people suffering from it. Future studies are necessary to improve the criteria for treating the complexity of this disease.

## Figures and Tables

**Figure 1 healthcare-11-00530-f001:**
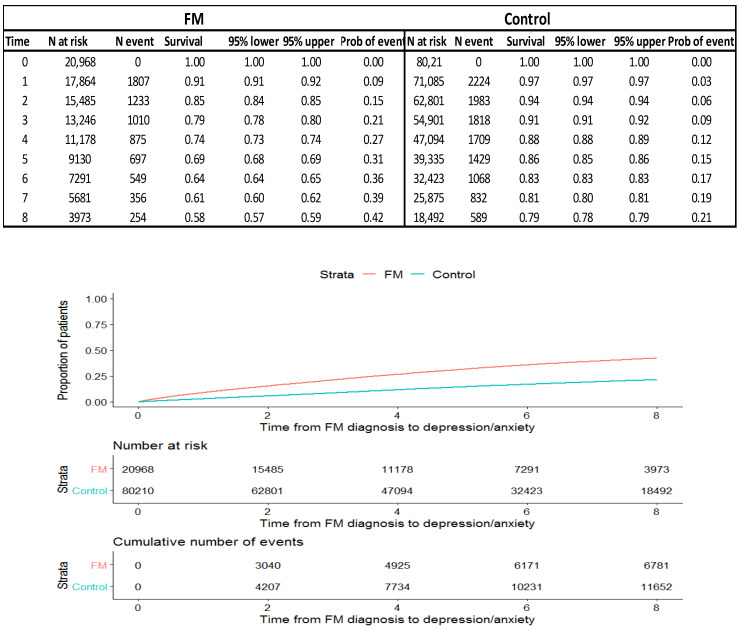
Kaplan–Meier survival curve for anxiety and/or depression from 2010–2017, separating patients according to the general population’s FM prevalent population and FM incidence population.

**Figure 2 healthcare-11-00530-f002:**
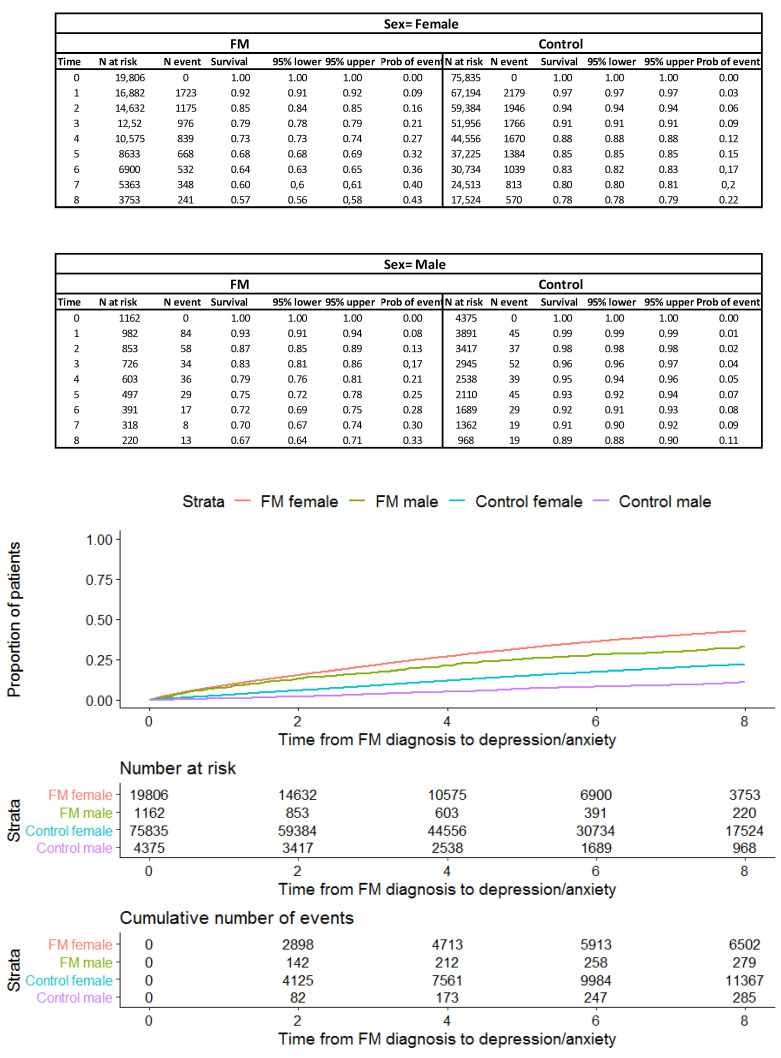
Kaplan–Meier survival curve at eight years for the outcome according to sex.

**Table 1 healthcare-11-00530-t001:** Demographic characteristics at baseline.

	Case—Incident	Case—Prevalent	*p*-Overall ^b^	Case, n (%)	Control, n (%)	*p*-Overall ^b^
	N = 13,327	N = 7641		N = 20,968	N = 80,210	
Sex			0.020			0.633
Female	12,551 (94.2%)	7255 (94.9%)		19,806 (94.5%)	75,835 (94.5%)	
Male	776 (5.82%)	386 (5.05%)		1162 (5.54%)	4375 (5.45%)	
Age ^a^	52.6 (12.1)	53.6 (11.9)	<0.001	53.0 (12.1);	52.6 (11.8);	<0.001
	52 [45; 60]	54 [46; 62]		53.0 [45.0;61.0]	53.0 [45.0;60.0]	
**MedeA ***			0.006			<0.001
Missing	9 (0.07%)	11 (0.14%)		20 (0.10%)	121 (0.15%)	
R	2296 (17.2%)	1304 (17.1%)		3600 (17.2%)	14,450 (18.0%)	
U	946 (7.10%)	641 (8.39%)		1587 (7.57%)	6270 (7.82%)	
U1	1734 (13.0%)	1015 (13.3%)		2749 (13.1%)	11,515 (14.4%)	
U2	1939 (14.5%)	1136 (14.9%)		3075 (14.7%)	12,039 (15.0%)	
U3	2201 (16.5%)	1214 (15.9%)		3415 (16.3%)	12,374 (15.4%)	
U4	2164 (16.2%)	1240 (16.2%)		3404 (16.2%)	12,238 (15.3%)	
U5	2038 (15.3%)	1080 (14.1%)		3118 (14.9%)	11,203 (14.0%)	
**Patients at the end of follow-up**			<0.001			<0.001
Alive	12,445 (93.4%)	6767 (88.6%)		19,212 (91.6%)	70,691 (88.1%)	
Dead	178 (1.34%)	202 (2.64%)		380 (1.81%)	1894 (2.36%)	
Transferred	704 (5.28%)	672 (8.79%)		1376 (6.56%)	7625 (9.51%)	
**Alcohol risk**:			<0.001			<0.001
Without risk	6918 (51.9%)	1768 (23.1%)		8686 (41.4%)	23,686 (29.5%)	
Low risk	1981 (14.9%)	367 (4.80%)		2348 (11.2%)	8250 (10.3%)	
High risk	66 (0.50%)	24 (0.31%)		90 (0.43%)	345 (0.43%)	
Unknown	4362 (32.7%)	5482 (71.7%)		9844 (46.9%)	47,929 (59.8%)	
**Tobacco consumption**:			<0.001			0.000
Non-smoker	7146 (53.6%)	3614 (47.3%)		10,760 (51.3%)	35,159 (43.8%)	
Smoker	2729 (20.5%)	1500 (19.6%)		4229 (20.2%)	12,033 (15.0%)	
Ex-smoker	1479 (11.1%)	526 (6.88%)		2005 (9.56%)	5712 (7.12%)	
Unknown	1973 (14.8%)	2001 (26.2%)		3974 (19.0%)	27,306 (34.0%)	

* MedeA categories: R rurality areas, U urban areas, and MedeA deprivation index into quartiles where U1: 1st quartile and U5: 5th quartiles are the least and most deprived urban areas, respectively. ^a^ mean (standard deviation), median [1st–3rd quantile] and *t*-test contrast. ^b^ the *p*-value of the chi- square test.

**Table 2 healthcare-11-00530-t002:** Prevalences and numbers of anxiety and depression diagnoses in patients with and without FM.

	Case—Incident	Case—Prevalent	*p*-Overall *
	N = 13,327	N = 7641	
	(n (%))	(n (%))	
**Anxiety and/or depression**	3750 (28.1%)	3031 (39.7%)	<0.001
**Anxiety post data index**	2409 (18.1%)	2014 (26.4%)	<0.001
No. of diagnoses of anxiety ^a^	0.19 (0.43)	0.29 (0.52)	<0.001
**Depression post data index**	2159 (16.2%)	1826 (23.9%)	<0.001
No. of diagnoses of depression ^a^	0.18 (0.42)	0.27 (0.51)	<0.001
**Case**		**Control**	***p*-Overall ^b^**
**N = 20968**		**N = 80210**	
**(n (%))**		**(n (%))**	
6781 (32.3%)		11,652 (14.5%)	0.000
4423 (21.1%)		8756 (10.9%)	0.000
0.23 (0.47)		0.12 (0.34)	0.000
3985 (19.0%)		4661 (5.81%)	0.000
0.21 (0.46)		0.06 (0.26)	0.000

^a^ Mean (standard deviation) and *t*-test contrast; * Chi-square test to compare incident case vs. prevalent case; ^b^ Chi-square test to compare case vs. control.

**Table 3 healthcare-11-00530-t003:** Cox model of time to onset of anxiety and/or depression with the variables group, sex, age and MedeA.

Variables	HR *	CI 95%	*p*-Value
**Group**			
Control	0.42	(0.40; 0.43)	<0.05
**Sex**			
Men	0.55	(0.50; 0.60)	<0.05
**Age**			
25–34	0.90	(0.76; 1.06)	0.21
35–44	0.94	(0.80; 1.10)	0.42
45–54	0.93	(0.79; 1.09)	0.38
55–64	0.90	(0.76; 1.05)	0.19
65–74	0.90	(0.76; 1.06)	0.19
≥75	1.04	(0.88; 1.24)	0.63
**MedeA**			
U	0.95	(0.90; 1.02)	0.16
U1	0.89	(0.84; 0.93)	<0.05
U2	0.96	(0.91; 1.01)	0.13
U3	0.96	(0.92; 1.01)	0.14
U4	1.01	(0.96; 1.06)	0.81
U5	1.02	(0.97; 1.07)	0.5

* HR: hazard ratio; CI: confidence interval.

## Data Availability

The data presented in this study are available upon request from the corresponding author.

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
