# Peer review of "Gender, Mental Health and Socioeconomic Differences in Fibromyalgia: A Retrospective Cohort Study Using Real-World Data from Catalonia"

_healthcare, 2023, doi:10.3390/healthcare11040530_

Round 1
Reviewer 1 Report
Review
Thank you for the opportunity to review the manuscript “Gender, Mental Health and Socioeconomic Differences in Fibromyalgia: a retrospective cohort study using real-world data from Catalonia.”
This article reports estimates risk of incident anxiety and depression in patients with and without fibromyalgia (FM). Data is from a medical records information system developed for research in primary care (SIDIAP), and other complementary data sources, in Catalonia with 168, 294 patients (56, 098 of them with FM paired with 112, 196 patients without FM on age, sex and primary care team). There is a lack of literature regarding comorbidities of fibromyalgia in population-based data. The strength of the study is its large cohort including also socioeconomic information. I have some comments and suggestions.
Major comments:
The introduction is informative and relevant but lacks argument as to why this study is needed. What is the knowledge gap?
The methods section needs to be more thorough and include more details about how the study was conducted, the different steps, and why different choices were made.
Specifically:
Is it incident or prevalent FM cases in this study?
Is the exposure, FM, time dependent? Do you have dates for when the diagnostic codes were registered?
From what time point is Kaplan Meier analyzed and how is that motivated, both for the FM patients and the controls?
It is not explained anywhere how the cases goes from N=56 098 to N=20 968 in Table 1, the same with controls that goes from N=112 196 to N=80 210. Please provide this information.
The index, MedeA needs to be described in more detail, it is stated that the index is in quintiles, but in Table 1 there is an empty space before the first row, then R, U and U1-U5. What does the empty space, R and U mean, please make this clearer.
The diagnostic code F49.1 is used for anxiety (row 91). Is this a mistake? F49.1 is not a diagnose in ICD-10. Do you maybe mean F41.9? In that case, what is the argument for not using the whole section of F41? Analog with using F32 for depression and not just code F32.9?
One sentence in method section state: “Diagnoses coded as dysthymia (F34.1) and postpartum depression (F53.0) were excluded.” Do you mean you excluded patients with ICD-10 code F34.1 and F53.0 at baseline? If so, what is the argument for excluding those diagnoses? Why did you for example not exclude code F33?
What does post data index mean in Table 2? This concept needs to be explained in the methods section.
In Table 3, do you mean that you have adjusted the model for confounding factors in your Cox regression? Please explain this in more detail in the methods section
Statistical analysis section needs more details. For example: What descriptive analysis were used?
Row 99-101: “In order to describe and compare the time to onset of anxiety and/or depression between different groups”. Referring to the question above: From what time point is Kaplan Meier analyzed, both for the FM patients and the controls?
Did you adjust for any confounders?
“Intervals”, do you mean Confidence intervals?
Minor:
The aim in the introduction and in the beginning of the discussion differs somewhat. Suggestion to use the same aim throughout the ms.
Tables 1 and 2 lack information on what is shown in the table in general e.g numbers, percentages, CI etc.
In Table 1 information in a footnote about MedeA’s different categories is suggested.
In the Discussion section the authors are suggested to add references to some statements:
Row 179: “Their onset worsens the progress of the disease and makes it become chronic”.
Row: 181: “These high levels of anxiety have also been observed in other studies. “
Row 200-201: “…some studies highlight a high frequency of psychiatric disorders in men with FM”
Row 187-189: These two sentences do not add up: “…women are at higher risk of anxiety and depression than men at diagnosis. This is consistent with the other studies reported in the literature in which FM is more prevalent in females”. First you state that mood disorders are more common among women in your study and then you say it’s consistent with FM being more prevalent among women. These are two different things. Please rephrase or add another reference.
Author Response
Thanks for your revision. We attach comments
Regards

Reviewer 2 Report
I thank the authors for the study. It is a very interesting study in a field that is really starting to take off. I have only one suggestion to improve the discussion.
I would suggest that you include studies from other countries as well. It seems very much Spain focused (I know that Spain has the most studies on it, but as studies have shown, there are differences in the impact of FM on mental health according to the region).
Van Overmeire R, Vesentini L, Vanclooster S, Muysewinkel E, Bilsen J. Sexual Desire, Depressive Symptoms and Medication Use Among Women With Fibromyalgia in Flanders. Sex Med. 2022 Feb;10(1):100457. doi: 10.1016/j.esxm.2021.100457. Epub 2021 Nov 25. PMID: 34839232; PMCID: PMC8847810.
Mutti GW, de Quadros M, Cremonez LP, Spricigo D, Skare T, Nisihara R. Fibromyalgia and sexual performance: a cross-sectional study in 726 Brazilian patients. Rheumatol Int. 2021 Aug;41(8):1471-1477. doi: 10.1007/s00296-021-04837-z. Epub 2021 Mar 16. PMID: 33725132.
There might be more studies, but at least this way you have a more northern country (Belgium) and a country on another continent (Brasil). I would just discuss it somewhat in the discussion, just to give context.
And here the study on the impact of FM:
Ruiz-Montero PJ, Segura-Jimenez V, Alvarez-Gallardo IC, Nijs J, Mannerkorpi K, Delgado-Fernandez M, van Wilgen CP. Fibromyalgia Impact Score in Women with Fibromyalgia Across Southern, Central, and Northern Areas of Europe. Pain Physician. 2019 Sep;22(5):E511-E516. PMID: 31561664.
Also, a couple of strange formulations. E.g. “FM is a disease that is associated with anxiety and depression although men are at lower risk of anxiety and depression at diagnosis”. It would read better if it was something like “FM is a disease that is associated with anxiety and depression. Women have a higher risk of anxiety and depression at diagnosis” or “FM is a disease that is associated with anxiety and depression. However, men have a lower risk …”
Also in the conclusion: “Future studies are necessary to discuss the better criteria to treat the complexity of the disease”, which is grammatically speaking odd. I imagine the authors meant “to discuss to better the criteria”. Or “to discuss to improve the criteria”.
I would suggest the author re-read the manuscript and adjust sentences that are quite difficult to understand.
Other than that, I think the study has a lot of merit.
Author Response

(The authors gave the same response as above.)

Round 2
Reviewer 1 Report
The authors have responded and have improved the manuscript. However, there are still a few major and minor remaining issues.
Abstract
The sentence beginning with “Survival from being diagnosed with anxiety and/or depression is lower in FM patients..” is difficult to understand. Could you phrase this in a way easier to understand?
This phrasing is also suggested to be changed in the Results section
Introduction
The sentence beginning with “Men also suffer from it,…” suggestion to instead write “Men also suffer from the disease,…”
Materials and Methods
One major issue concerns prevalent vs incident cases. The authors have given information about how the study include both. This is crucial. Therefore, explain already in the Materials and Methods section that patients with FM are a mix of prevalent cases (treated as if they got their diagnosis on 1/1 2010) and incident cases. How many were prevalent and incident cases respectively? Moreover, please motivate why these groups were analyzed together. Would for example a sensitivity analyses were the groups were analyzed separately provide useful information about how the risk of anxiety/depression differed between prevalent and incident cases?
What does the word “mortality” mean in this sentence?: “The demographic variables studied were sex, age and information on socio-economic status recorded through mortality in small Spanish areas…”
Please add information about size of study population in adjunction to the sentence “Both groups of participants were free of anxiety and depression at baseline (2010).”
It is still unclear to me what is the argument for only using F41.9 and not using the whole section of F41, since you use the whole section F32 for depression and not just code F32.9?
Statistical analysis:
In the sentence finishing with ”…the Kaplan Meier was calculated as of 01/01/2010 for prevalent cases and for incident cases it was calculated on actual diagnostic data.” I wonder if the last word should be “dates”?
Results
In relation to my previous comments (in the Materials and Methods section) the results section might potentially need be updated.
This sentence is incomplete: “The demographic and baseline characteristics are described in…”
In Table 2, motivate and explain the use of mean and median. Figure legends differ between in the Table and below the Table.
Discussion
In relation to my previous comments (Materials and Methods and results sections) the discussion section might potentially need be updated.
The following text need some revision to be understandable: “Anxiety was 21.1%, twice higher than in the control group, and depression was 19%, more than three times higher than in the control group. These results coincide with some studies comparing FM patients with healthy control groups, and the prevalence is significantly higher in mental disorders.”
The last part of this sentence could benefit from some further explanation, it is now unclear: ”When analysing by sex, the percentatge of women with depression and anxiety was reported to be higher than that of men, this may be due to the fact that men have a lower rate of diagnosis which has hindered studies on gender differences (29 ) and some studies highlight a high frequency of psychiatric disorders in men with FM (30), which is contrary to what happen in the general population, where women present depression and anxiety disorders more frequently than men”
Please problematize the fact that cases are a mix of prevalent and incident FM. For example the “time from FM” in Figure 1 and 2 is problematic concering prevalent cases, some might have had FM for several years before starting date of the study.
Author Response
we attach the document
Best regards

Round 3
Reviewer 1 Report
The authors have answered most questions and improved the manuscript accordingly.
The Abstract section is missing in the Revised version.
The expression “Survival after being diagnosed with anxiety and/or depression is lower in FM patients..” is difficult to comprehend.
The authors have explained why prevalent and incident cases were analyzed together. However, there was no response to the suggestion to perform a sensitivity analysis where the groups were analyzed separately. This kind of analysis would show if the results are valid for both groups.
No of diagnoses… in Table 2 is still unclear to me and not explained in the text.
Author Response
Thank you for your contributions. We have attached all the checks you asked for us.
Regards
